# A Smart Active Phase-Change Micropump Based on CMOS-MEMS Technology

**DOI:** 10.3390/s23115207

**Published:** 2023-05-30

**Authors:** Wenzui Jin, Yimin Guan, Qiushi Wang, Peng Huang, Qin Zhou, Kun Wang, Demeng Liu

**Affiliations:** 1School of Microelectronics, Shanghai University, Shanghai 201800, China; wenzui_jin@shu.edu.cn (W.J.); qiushiwang@shu.edu.cn (Q.W.); zhouqin@shu.edu.cn (Q.Z.); 2Shanghai Aure Technology Limited Company, Shanghai 201800, China; yimin_guan@shu.edu.cn (Y.G.); peng.huang@aurefluidics.com (P.H.); kun.wang@aurefluidics.com (K.W.)

**Keywords:** phase-change micropump, active micropump, CMOS-MEMS

## Abstract

The rational integration of many microfluidic chips and micropumps remains challenging. Due to the integration of the control system and sensors in active micropumps, they have unique advantages over passive micropumps when integrated into microfluidic chips. An active phase-change micropump based on complementary metal–oxide–semiconductor–microelectromechanical system (CMOS-MEMS) technology was fabricated and studied theoretically and experimentally. The micropump structure is simple and consists of a microchannel, a series of heater elements along the microchannel, an on-chip control system, and sensors. A simplified model was established to analyze the pumping effect of the traveling phase transition in the microchannel. The relationship between pumping conditions and flow rate was examined. Based on the experimental results, the maximum flow rate of the active phase-change micropump at room temperature is 22 µL/min, and long-term stable operation can be achieved by optimizing heating conditions.

## 1. Introduction

Over the past two decades, microfluidics has greatly benefited fundamental and applied research in different fields, such as medicine, biology, and chemistry [1]. Microfluidic devices achieve designed functionalities by utilizing some unique characteristics of microscopic flows, such as the impact of physics of scale on the balance of forces, reduced diffusion timescale, and enhanced surface area per volume [2]. As an executive device, the micropump plays an essential role in microfluidic devices, including in chemical analysis systems, micromixers, and microdosing systems.

Micropumps have a large variety of working mechanisms. From the point of view of the operating principle, micropumps are divided into two categories: pumps with mechanical moving parts and those without such parts. Mechanical moving parts are often delicate components that must be carefully manufactured and installed for pump reliability. With pumps being increasingly smaller, complications in macroscale pumps are often magnified in microscale pumps. Therefore, more attention has been paid to micropumps without mechanical moving parts. Micropumps without mechanical moving parts are generally valveless and take advantage of the specified properties of fluids. Several types of these micropumps have been fabricated successfully, such as phase-change micropumps [3], electrohydrodynamic pumps [4], electroosmotic pumps [5], magnetohydrodynamic pumps [6], and electrowetting pumps. Phase-change micropumps have some outstanding characteristics, such as high miniaturization potential, simple structure, and low cost, as shown in Table 1. Different from traditional micropumps, the maximum pumping pressure and flow rate are more considerable in microdevices when compared to their pump size.

The phase-change micropump is a novel micropump initially presented by Ozaki et al. in 1992 [3]. Its structure is simple and consists of a microchannel and a series of heater elements along the microchannel. Due to the thermal properties of the fluid used, the periodic opening and closing of the heater produce periodic phase changes in the fluid. This is due to the pumping effect caused by the evaporation and condensation of the fluid in the capillary tube and the difference between the viscous forces of the fluid and vapor [8]. As a result, the fluid is pumped in the scanning direction of the heater.

Research on phase-change pumps has been scarce in the past 20 years, and most previous studies have remained at the millimeter level. With the rise of the lab-on-chip concept, micropumps with higher integration and faster flow rates have attracted academic attention. In this study, we designed a smart phase-change micropump using the complementary metal–oxide–semiconductor–microelectromechanical system (CMOS-MEMS) platform. Unlike traditional phase-change micropumps, a smart micropump has some outstanding characteristics, such as an on-chip control system, small size, high precision, convenient use, and intelligence. In the future, real-time monitoring of the flow rate of smart micropumps can be achieved by equipping a flow sensor [9,10] at the outlet, and the flow rate can be adjusted according to the requirements.

## 2. Materials and Methods

### 2.1. Principle of Pumping

A simplified physical model was established in this study. Generally, due to the small diameter of micropumps and the slow fluid velocity inside their tube, the fluid flow inside the tube can be simplified as a laminar flow model. The small diameter of the tube also sharply reduces the thermal time constant of the fluid flowing in the microchannel, allowing the liquid to undergo frequent phase changes in a short time [11]. When a liquid evaporates into vapor, kinematic viscosity is significantly reduced. Due to the significant difference in kinematic viscosities of the liquid and vapor phases, phase-change micropumps can work stably, as described below. As shown in Figure 1, the pumping process of a micropump is divided into three steps.

In Step I, no vaporized areas exist in the channel. The pressure gradient is constant throughout the microchannel, and the fluid flows from outlet O to inlet I.

In Step II, only the first heater is electrified; if the voltage is high enough, the liquid above the heater will vaporize. As the bubble continues to grow, it almost fills the cross-section and begins to grow along the micropump axis [12,13]. If a vaporized region exists in the microchannel, most of the total pressure drop occurs in the vaporized area because of the sealing effect [3]. The fluid still flows from outlet O to inlet I, but the mass flow rate is much lower than in Step I.

In Step III, when the next heater is turned on, a vaporized area moves along the scanning direction of the heat source. The bubble is in a stage of asymmetric heating, with one end in the condensation phase and the other in the vaporization phase. As the vapor pressure is strongly positively correlated with the temperature difference [14], asymmetric temperature gradients lead to asymmetric pressure gradients. If the moving velocity of the phase boundary (E, C) is large enough, the pressure drop in the vaporized area becomes bigger than the pressure difference between the inlet and the outlet, as shown in Figure 2. The liquid starts flowing from the inlet to the outlet, indicating the pumping effect of a phase-change pump type. This pumping mechanism is attributed to the large difference in the kinematic viscosities between the liquid and gas phases of the fluid. The micropump can ultimately pump the liquid from the inlet to the outlet by repeating these three steps.

To further explore the pumping mechanism of the micropump, the condition of pumping is examined via theoretical analysis as follows. For an incompressible, laminar, and viscous liquid flow in a microchannel, the relationship between the pressure drop and the coefficient of viscosity can be obtained from Poiseuille’s law:(1)$$∆P=\frac{Q\times 8\eta L}{\pi R^{4}},$$
where L is the distance traveled by the fluid, η is the coefficient of viscosity of the fluid, Q is the volume flow rate of the fluid, ΔP is the pressure drop, and R is the diameter of the microchannel. The following equation is used to calculate the pressure drop:(2)$$∆P=\frac{m\mu L}{F},$$
where *μ* is the kinematic viscosity of the fluid, *m* is the mass flow rate, and *F* = πr^4^/8.
(3)m=Qρ=SUρ,
where *U* is the moving velocity of the vaporized area and *ρ_L_* and *ρ_V_* are the densities of the liquid and vapor phases, respectively. Since the kinematic viscosity of the liquid phase is much larger than that of the gas phase, it is assumed that *ρ_V_* << *ρ_L_*.
(4)$$mo-mc≅SU\rho_{L}\;\;mc-mi≅-SU\rho_{L},$$

From Equation (4), the mass flow rate *m*(=*mo* = *mi*) is
(5)$$m=\frac{-∆P_{T}F+SUL_{C}\mu_{V}}{L_{T}\mu_{L}+L_{C}(\mu_{V}-\mu_{L})},$$

Here, *L_T_* = *L_O_* + *L_C_* + *L_I_* and Δ*P_T_* = *P_I_*-*P_O_*. In Steps I–II, *U* = 0, and the phenomenon of liquid flowing from outlet O to inlet I is called the free leakage flow.

The condition for pumping is as follows:(6)$$-UL_{C}>\frac{-∆P_{T}F}{S\rho_{L}\mu_{V}},$$

From the above equation, the condition for pumping in a micropump is that *U* is negative. In the step where the microchannel size is determined, the moving velocity and the length of the vaporized area determine the effectiveness of a single pump.

### 2.2. Micropump CMOS Design

The CMOS circuit consists of a 24-bit shift register, a temperature sensor, a power-on reset circuit (POR), and a set of metal–oxide–semiconductor field-effect transistors (MOSFETs) for controlling the heater. The chip has only two logical control signals, UPDIRECT and CLK. The UPDIRECT signal controls the flow direction and determines the scanning direction of heating, whether in a sequential order (1–24) or a reverse order (24–1). As shown in Figure 3, the CLK signal is divided into “fire” and “data” signals. The “data” signal is converted into “data_1” to “data_24” through the 24-bit shift register.

The resistor is connected between the high-voltage power supply and the drain of the low drain N-channel MOSFET (LDNMOS) power transistor on the low side. The chip has a microchannel with 24 heating units. The fire signal and the data signal are logically ANDed together to generate the final control signal, which is used to control the switching state of the LDNMOS. Figure 3b shows the driving circuit of a single heater.

Due to the chip’s small size, only one set of temperature sensors is equipped in the design for real-time temperature monitoring. As shown in Figure 4, a group of temperature sensors contains two temperature-sensitive units, which convert the ΔVbe proportional to the absolute temperature into a proportional to the absolute temperature (PTAT) current signal through a resistor [15,16]. The voltage of the bipolar transistor is shown in Equation (7).
(7)$$V_{BE}=\frac{KT}{q}\ln\left(\frac{I_{C}}{I_{S}}\right),$$
where *K* is the Boltzmann constant; *q* is the electron charge; *T* is the temperature in Kelvin; *I_C_* and *I_S_* are the collector and saturation currents of the PNP, respectively; *m* is the bipolar transistors’ collector current density ratio; and *n* is the bipolar transistors’ ratio. Since the current mirror provides current for the bipolar transistors, M1 and M2, and ensures that the source potential of both transistors is the same, the voltage across the resistor is calculated as shown in Equation (8):(8)$${∆V}_{BE}=V_{BE1}-V_{BE2}=\frac{KT}{q}\ln\left(mn\right),$$
(9)$$I_{PTAT}=\frac{{∆V}_{BE}}{R},$$

According to the selected *m* and *n*, a voltage Δ*Vbe* proportional to the absolute temperature will be generated. This voltage will be converted into a current signal proportional to the temperature (*I_PTAT_*) through a resistor and outputted as Ts_out to the external circuit of the chip.

In this chip, a power-on reset circuit is designed to automatically trigger a reset signal upon power-up, thereby resetting the entire chip. The reset signal is automatically turned off after the reset is completed.

### 2.3. Micropump MEMS Design

The structure of the micropump chip is shown in Figure 5a. The micropump consists of a microchannel and a series of heater elements along the microchannel. According to the description, liquid enters the microchannel through the inlet and is heated sequentially by 24 heaters before being pumped out from the outlet. As shown in Figure 5b,c, each heater has a size of 43 × 18 µm, a resistance value of 1 kΩ, and a distance of 24.3 µm between adjacent heaters. The width of the microchannel is 60 µm, the length is 1470 µm, and the height is 20 µm. A 5.4 mm × 3 mm active phase-change micropump is fabricated using the CMOS-MEMS platform.

For the stable growth of bubbles, the position of the first heater is crucial for the operation of the phase-change micropump. On the one hand, if the first heater is located too close to the inlet, bubbles may overflow during the growth process. On the other hand, if the first heater is located too far away from the inlet, the number of heaters that can be integrated into the microchannel will decrease. Therefore, the length of the microchannel was selected as 1470 µm, and the position of the first heater was set at 250 µm from the inlet, resulting in a ratio of 0.83 for the distance from the first heater to the inlet to the length of the microchannel [17]. In this scheme, a vapor bubble is mostly contained within the microchannel during its lifetime.

Academic research on the design of active phase-change micropumps has been limited. This paper presents the design and development of an active phase-change micropump based on the CMOS-MEMS platform. Integrating microactuators and sensors allowed us to construct an active phase-change micropump design that precisely controls the fluid flow rate, as shown in Table 2. Compared to traditional phase-change micropumps, this study achieved a transition from macroscale to microscale pumping, which has significant implications for microfluidics. The control system of the active phase-change micropump is integrated into the chip based on the CMOS-MEMS platform, making it more conducive to chip integration and application. Our research results demonstrate that the CMOS-MEMS-based active phase-change micropump is a promising alternative to traditional phase-change micropumps, with the potential for extensive application in various fields, including biomedical engineering, chemical engineering, and environmental science. Moreover, this technology can create multiple channels on a chip, thus achieving high-flow-rate microfluidic operations.

### 2.4. Micropump Fabrication

Based on the CMOS-MEMS design platform, a schematic diagram of the CMOS-MEMS process is shown in Figure 6. The CMOS process is responsible for the logic control of the chip, including controlling and processing the input electrical signals to achieve functions such as heating the heating unit of the control chip, timing, and temperature reading. The subsequent MEMS process completes the fabrication steps, including the steps in the heating unit, the flow microchannel, the inlet and outlet ports, and the metal interconnections.

During MEMS fabrication, it is necessary to produce this portion through deep reactive ion etching (DRIE). As there is a hollow structure at the bottom of the MEMS after formation, a photosensitive polymer film (dry film) is used to produce the MEMS portion, as shown in Figure 6. The process involves attaching the dry film, followed by exposure and development using a photolithography machine, and, finally, high-temperature curing. After completing the first layer of channel fabrication, the silicon substrate is etched using silicon vias technology. Finally, a layer of dry film is used for capping and high-temperature curing.

Combining the CMOS-MEMS technology can bring many advantages, as follows:A high signal-to-noise ratio is possible while reducing the interference between external leads and sensitive positions and minimizing parasitic and cross-coupling effects.By integrating internal sensors and control signals, it is possible to achieve the integration of large array units and effectively reduce the number of external leads in these large arrays. This enables better integration of even larger arrays and facilitates the realization of an intelligent system.

### 2.5. Experimental System

An experimental set-up was built for the experimental investigations of the phase-change micropump described in the previous section, as schematically shown in Figure 7. The experimental apparatus consisted of a logic analyzer, soft tubes, and a DC power source. Ultrapure water was used as a working fluid.

After bonding with the ceramic substrate, the chip is connected to the soft tube through a 1 mm diameter steel tube. The logic analyzer provides the chip with clock and flow direction signals, while the DC power supply provides a high voltage of 35 V and a low voltage of 5 V. The length of the liquid moving inside the tube is a criterion for evaluating the average flow rate.

## 3. Results

### 3.1. Bubble Nucleation

The printing frequency of the CLK signal shown in Figure 8 is the frequency at which the 24 heating units are sequentially pumped once. During the heating process of a single heater, the CLK signal is divided into two parts, TH and TD. TH is the duration of the high voltage of 35 V being applied to the heater, and TD is the interval between the heating time of adjacent heaters. The UPDIRECT signal controls the flow direction of the fluid, and it enables a signal that precedes the CLK signal.

For a single heater, the obtained energy during one pumping cycle is represented by Equation (9):
(10)$$W=\frac{U^{2}TH}{R},$$
where U is the pulse voltage applied to the electrodes and R represents the heating resistance value.

Based on the trend of water’s nucleation rate with temperature variation, it can be inferred that the temperature required for liquid nucleation is approximately 315–360 °C [20]. As shown in Figure 9, the temperature on the heater is simulated according to Equation (9). With longer TH time, the temperature of the heater gradually increases. If TH ≥ 600 ns, the temperature of the heater reaches the nucleation temperature, enabling the generation of stable bubbles.

### 3.2. Optimization of Printing Frequency

According to the previous section, the effectiveness of a single pump depends on the size of the vaporized area and the moving velocity of the vaporized area. The size of the vaporized area is proportional to the TH time, and the moving velocity of the vaporized area is reflected in the TD time. The printing frequency determines the amount of liquid pumped per unit of time.

Under the given test conditions of TH = 800 ns and TD = 650 ns, the impact of the printing frequency on the average flow rate was investigated. The measured average flow rate and the circuit temperature for various printing frequencies are presented in Figure 10. The experimental results indicate that as the printing frequency increases, the average flow rate for achieving the maximum flow rate becomes shorter. This is mainly attributed to the fact that a higher temperature leads to a slower bubble collapse. The presence of residual bubbles at the inlet may cause bubble accumulation due to the temperature effects, as shown in Figure 11. When a bubble reaches a sufficient size, it is drawn into the micropump and blocks it, resulting in a lack of liquid vaporization above the heater and causing the micropump to stop working. Similarly, if bubbles accumulate at the outlet, the liquid is difficult to pump forward, and the micropump will also stop working. Notably, the micropump can remove residual bubbles above the heater and work normally.

### 3.3. Thermal Effect

Based on the experimental comparisons, the active micropump has a significant advantage over a previously published passive micropump [21] in terms of heat loss. The simulation results in Figure 12 illustrate the temperature variation of the silicon chip when heaters with a thickness of 1.5 μm and 3 μm receive the same amount of energy input. It is observed that the thin insulating layer of the silicon wafer has a higher temperature during heating, indicating that it absorbs more heat and, therefore, has a higher heat loss rate. In addition, the higher the temperature of the chip, the less energy is required for heater nucleation. The thin insulating layer of the silicon wafer cools down faster after heating has been stopped, indicating that the temperature loss rate of the chip is faster and not conducive to the subsequent heating of the heater.

As shown in Table 3, the improved thermal insulation performance of the active micropump has an impact on the flow rate. At the same flow rate, the coefficient of variation in the flow rate of the active micropump is reduced by 50% compared to the passive micropump. Moreover, at the same flow rate, the active micropump requires a lower frequency, has a lower chip temperature, and has a lower probability of bubble accumulation at both the inlet and outlet.

### 3.4. Optimization of Pumping Conditions

An optimal printing frequency was obtained based on the experimental results described above. The effects of changing TH time were examined, and the results are shown in Figure 13a,b. The vapor bubble strength can be changed during an experiment by changing the electrical pulse delivered to the resistor. The TH determines the size of the vaporized area. The average flow rate rises from 8.2 µL/min at TH = 600 ns to the maximum value at TH = 800 ns, and then the average flow rate decreases. The average flow rate continues to increase because the bubble size on the resistor increases. This behavior stops when the liquid is completely vaporized above the heater and the chip’s temperature keeps rising. A higher chip temperature results in slower bubble collapse, which is unfavorable for the operation of the micropump.

In the case of a printing frequency of 7 kHz, the changes in TH time have much less effect on the average flow rate than TD variations. The TD determines the moving velocity of the vaporized area. Figure 13c,d show the average flow rate as a function of TD. For TD < 350 ns, the average flow rate becomes rapidly more and more stable, indicating that two adjacent bubbles are in a stable contact state. The maximum effect is reached when TD = 350 ns. When TD equals 450 ns, the average flow rate decreases, indicating that two adjacent bubbles are approaching stable contact. The average flow rate drops in the range of 550 ns < TD < 650 ns, suggesting that the contact between adjacent bubbles is unstable—the instability of bubble contact results in a slower flow rate in the micropump. When TD exceeds 650 ns, the average flow rate drops abruptly and gradually approaches zero. This indicates that the moving velocity of the vaporized area is too slow, thus making it difficult for bubbles to make contact and drive the liquid.

### 3.5. Optimization of Pumping Stability

As a novel microfluidic component, stability testing is necessary to integrate a micropump. Stability testing of the micropump was carried out under the optimal pumping conditions obtained in the previous section. As shown in Figure 14, the flow rate of the micropump increases with time and reaches a stable value. After stable pumping for a period of time, the micropump stops working. This can mainly be attributed to the accumulation of bubbles due to higher temperatures, which gather at the inlet and outlet over time. When the bubbles become large enough, the micropump stops working. The randomness of bubble retention at high temperatures results in extreme instability in the flow rate and affects the time of cessation of the micropump.

According to the above analysis, it is vital to prevent the accumulation of bubbles at the inlet and outlet. Temperature directly affects the possibility of bubble accumulation. During the pumping process, the heater that is heated first serves as a preheating unit for the heater that is heated later. Based on the simulation results in the previous section, temperature is transmitted to the silicon in tens of microseconds, and the heater time of a single heater is on the order of microseconds. When temperature increases, the energy required for heater nucleation decreases, and the energy acquired by a heater corresponds to the duration of its heating time. The increase in chip temperature means that the energy required for nucleation decreases, and the excess energy is absorbed by the silicon substrate to increase the chip temperature. If all the heaters use the same heating conditions, the heat loss of the heater heated along the pumping direction gradually increases, which exacerbates the possibility of bubble accumulation.

As shown in Figure 15, the experimental results are presented for four groups of heaters with pulse durations of 800 ns, 700 ns, 600 ns, and 500 ns, respectively. The optimization of heating conditions reduces the coefficient of variation in the flow rate from 13% under constant heating conditions to 3%, as shown in Table 4. The coefficient of variation reflects the stability of the micropump in different experimental batches. By reducing the power loss of the heaters in the latter group, the possibility of bubble accumulation at the outlet is reduced, and the stable operating time of the micropump under variable heating conditions significantly increases compared to that under constant heating conditions. While achieving higher stability, the optimized heating conditions also result in a slight decrease in flow rate. In biology, stable long-term operation is crucial for micropumps, such as organ chip applications that require long-term perfusion [22,23]. It is unreasonable to integrate micropumps that cannot operate for a long time into microfluidic systems. Therefore, sacrificing the micropump flow rate for long-term stable operation is a desirable trade-off.

## 4. Conclusions

We fabricated a phase-change micropump using the CMOS-MEMS technique. A phase-change pumping principle of fluids in a microchannel was proposed, analyzed, and demonstrated. The effectiveness of this pumping mechanism is attributed to the significant difference in the kinematic viscosities between the liquid and gas phases of the fluid used, with asymmetric heating resulting in asymmetric vapor pressure. The advantages of the active phase-change micropump design are miniaturization and intelligence. By utilizing the CMOS-MEMS fabrication process, the sensor and control system are integrated on-chip, enabling the phase-change micropump to be scaled down from the macroscopic level to the microscopic level, thereby filling the gap in the field of phase-change micropumps.

A CMOS-MEMS platform was utilized to fabricate an active-phase change micropump with dimensions of 5.4 mm × 3 mm. The microchannel has a width of 60 µm, a length of 1470 µm, and a height of 20 µm. Based on the experimental results, the maximum flow rate of the active phase-change micropump at room temperature is 22 µL/min, and long-term stable operation can be achieved by optimizing heating conditions.

## Figures and Tables

**Figure 1 sensors-23-05207-f001:**
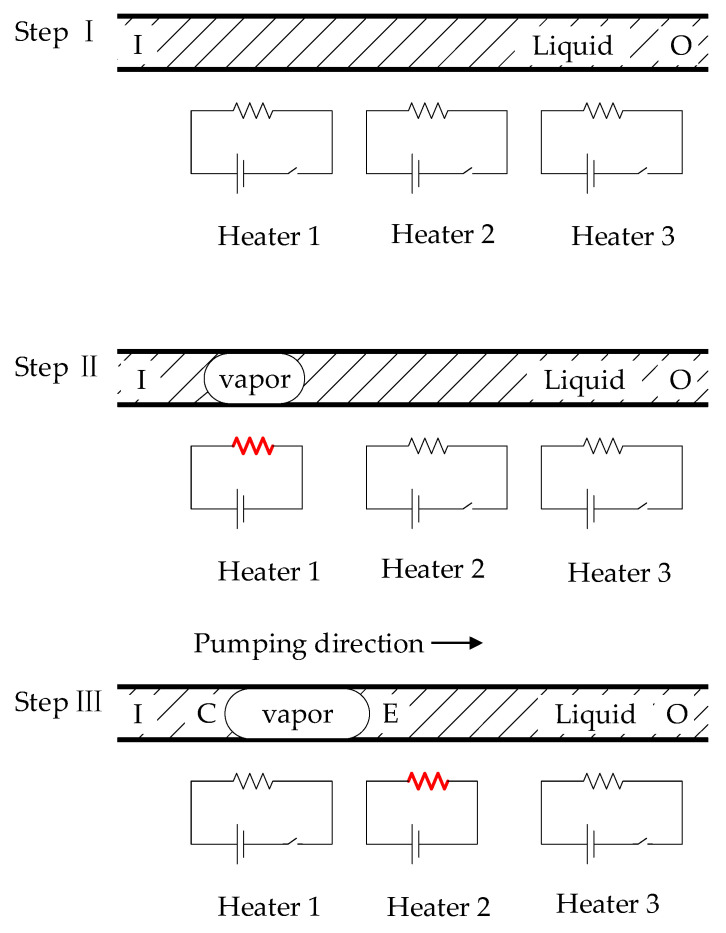
Illustration of the pumping mechanism.

**Figure 2 sensors-23-05207-f002:**
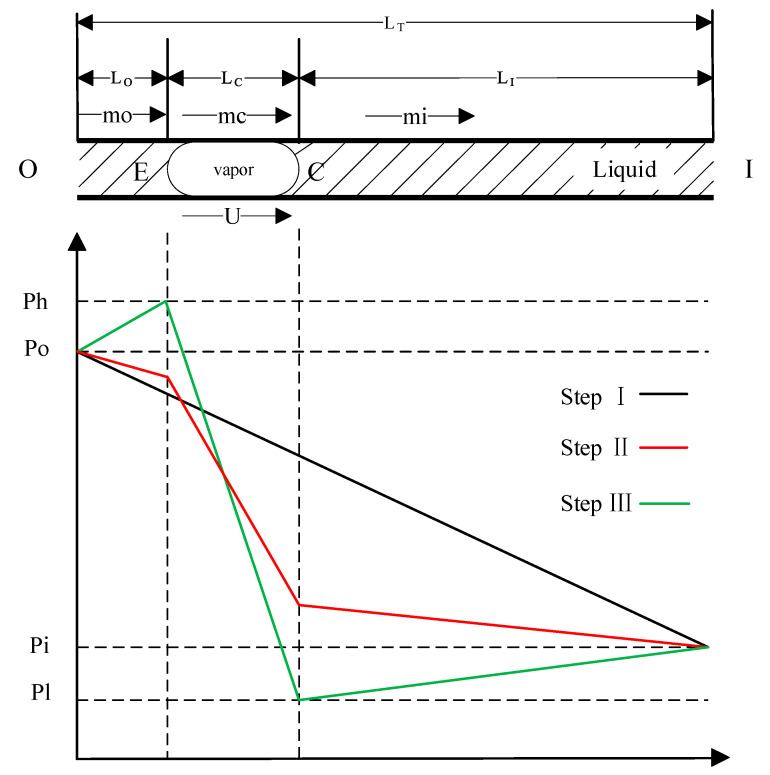
Pressure distribution in flow through a microchannel.

**Figure 3 sensors-23-05207-f003:**
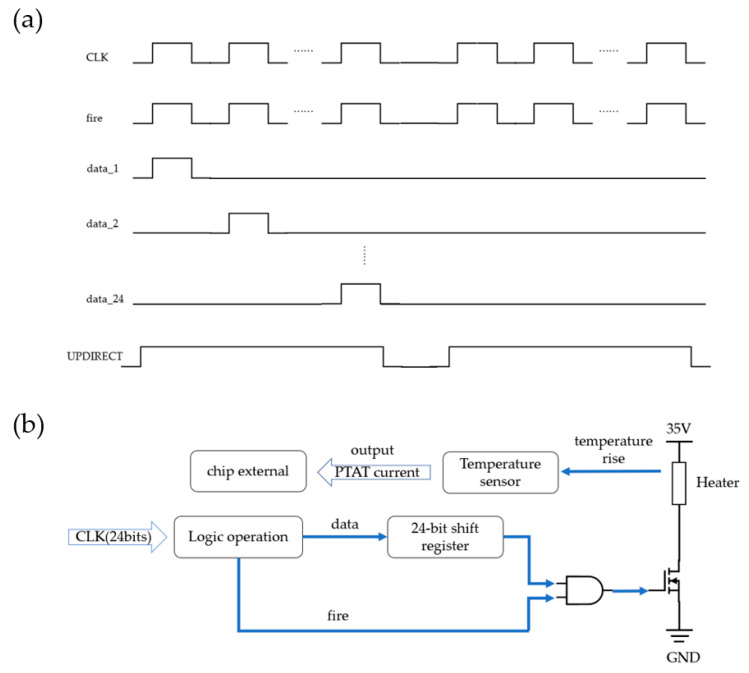
(**a**) Control logic sequence diagram and (**b**) the driving circuit of a single heater.

**Figure 4 sensors-23-05207-f004:**
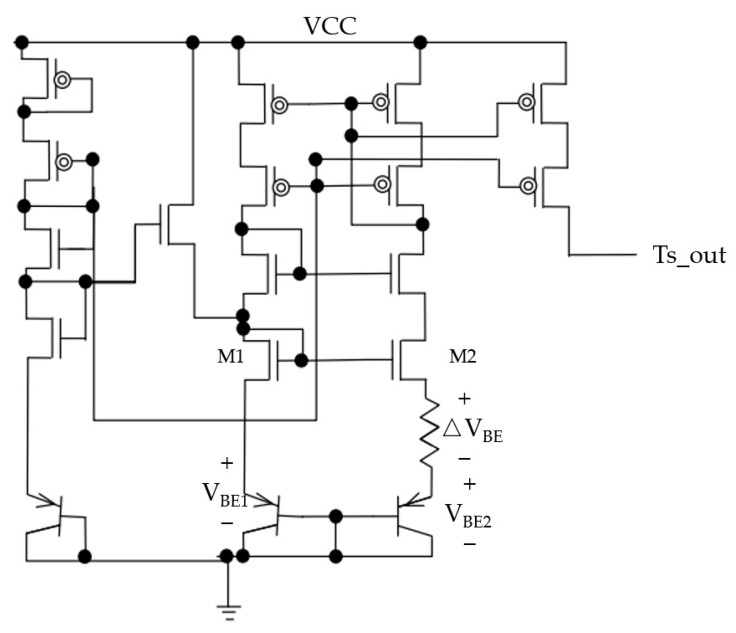
Topology of temperature sensors.

**Figure 5 sensors-23-05207-f005:**
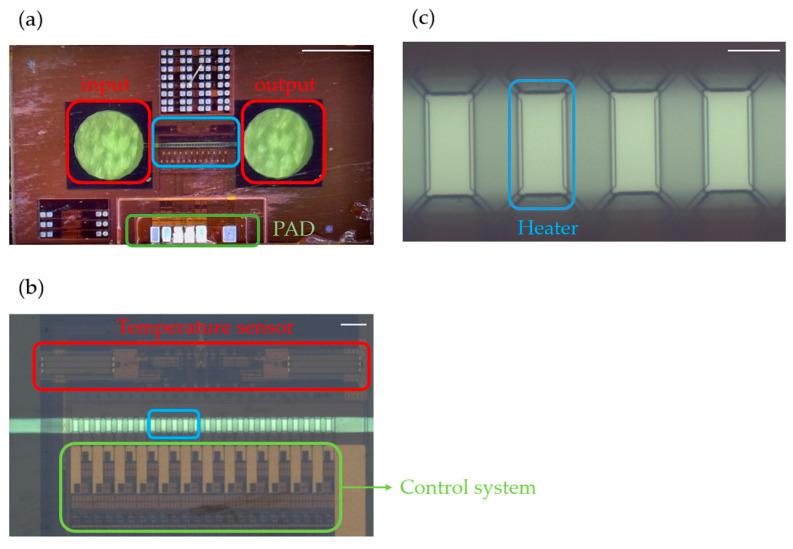
(**a**) Optical photo of the micropump (scale bar: 1 mm); (**b**) an amplified photo of the microchannel (scale bar: 100 µm); and (**c**) an amplified photo of the heater (scale bar: 20 µm).

**Figure 6 sensors-23-05207-f006:**
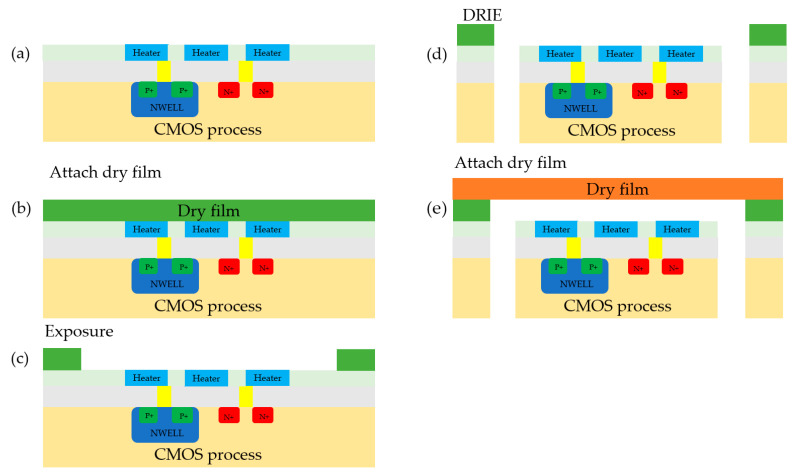
Fabrication of the micropump: (**a**) a chip fabricated using CMOS processing; (**b**) a chip with the first layer of dry film attached; (**c**) chip after exposure; (**d**) Chip after deep reactive ion etching; (**e**) a chip with the second layer of dry film attached.

**Figure 7 sensors-23-05207-f007:**
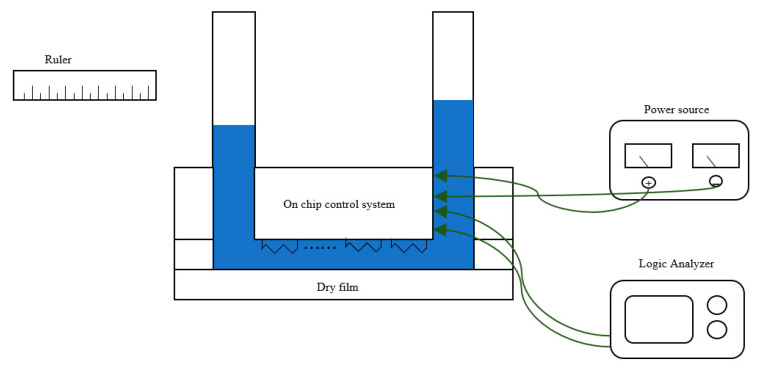
Experimental apparatus.

**Figure 8 sensors-23-05207-f008:**
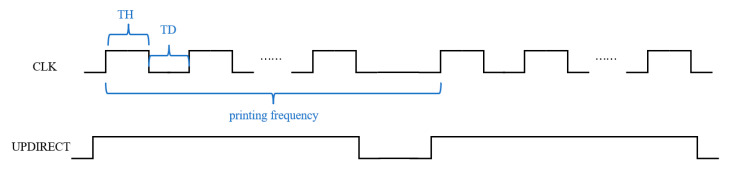
Micropump control signal timing.

**Figure 9 sensors-23-05207-f009:**
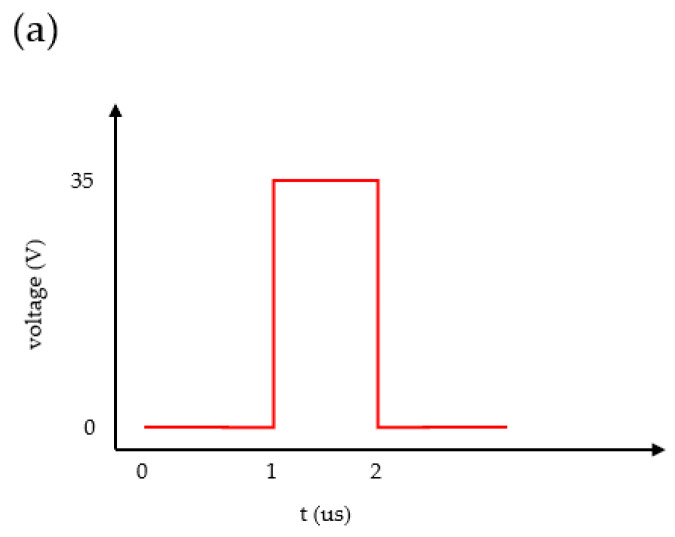
Bubble nucleation simulation: (**a**) the driving waveform imposed on the heater and (**b**) the temperature of the heater surface.

**Figure 10 sensors-23-05207-f010:**
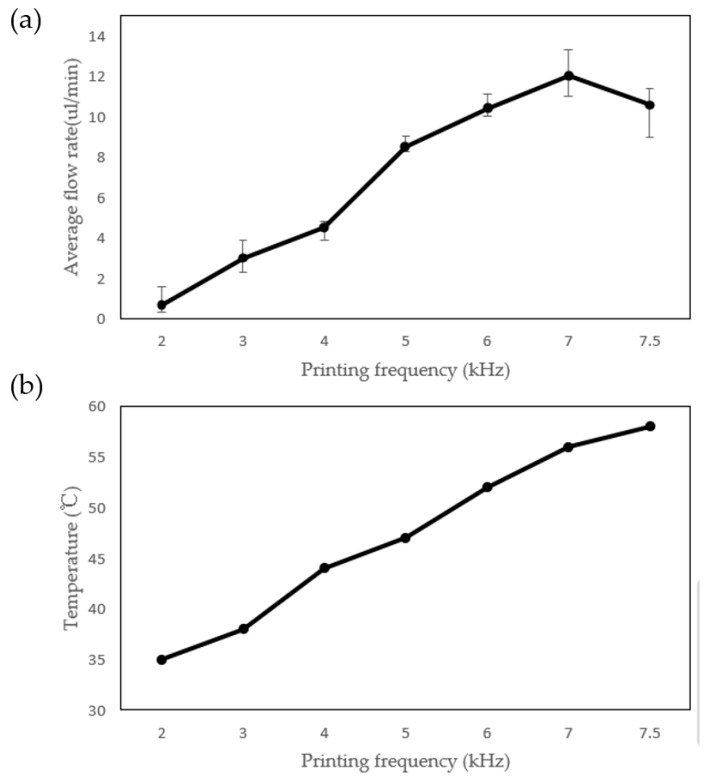
(**a**) The effects of printing frequency on the average flow rate and (**b**) the relationship between printing frequency and temperature.

**Figure 11 sensors-23-05207-f011:**
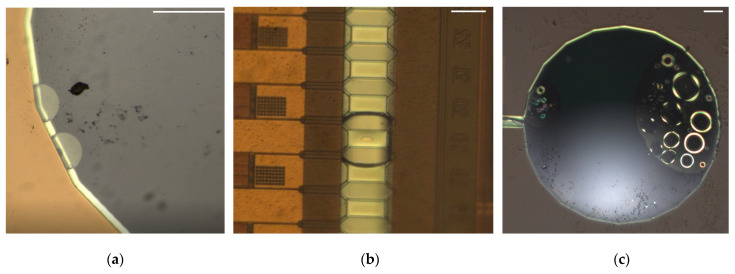
(**a**) Bubbles at the inlet (scale bar: 0.1 mm); (**b**) bubbles in the microchannel (scale bar: 43 µm); and (**c**) bubbles at the outlet (scale bar: 0.1 mm).

**Figure 12 sensors-23-05207-f012:**
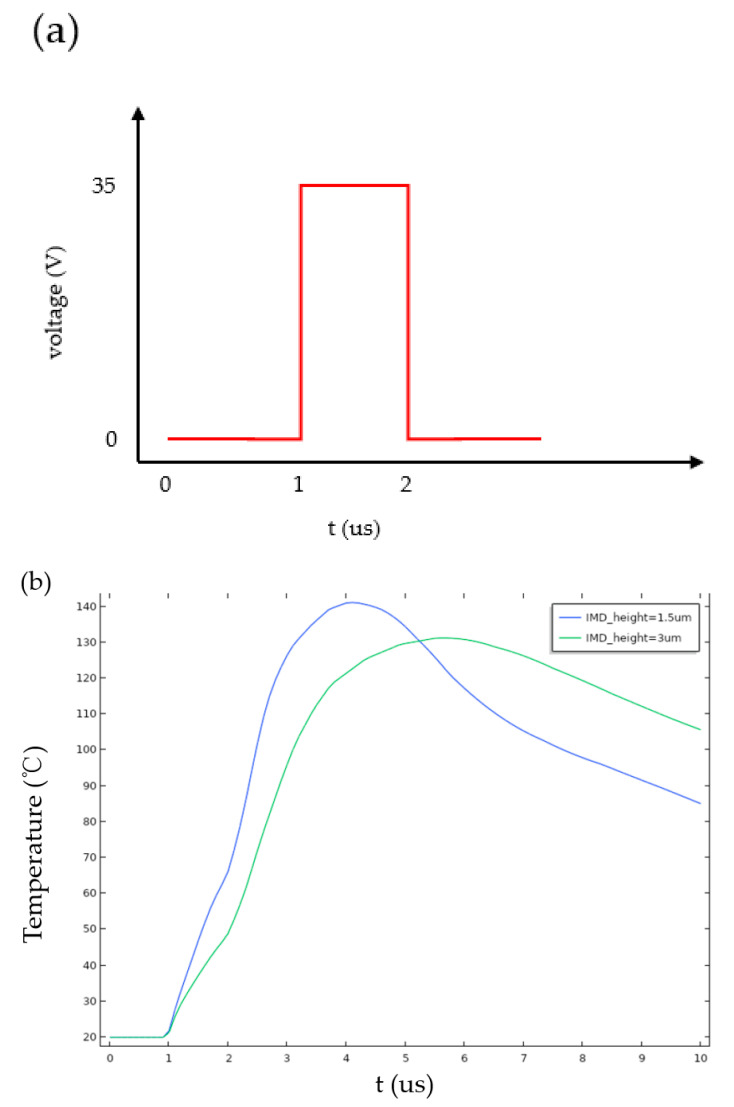
Thermal effect simulation: (**a**) the driving waveform imposed on the heater and (**b**) the temperature of the silicon.

**Figure 13 sensors-23-05207-f013:**
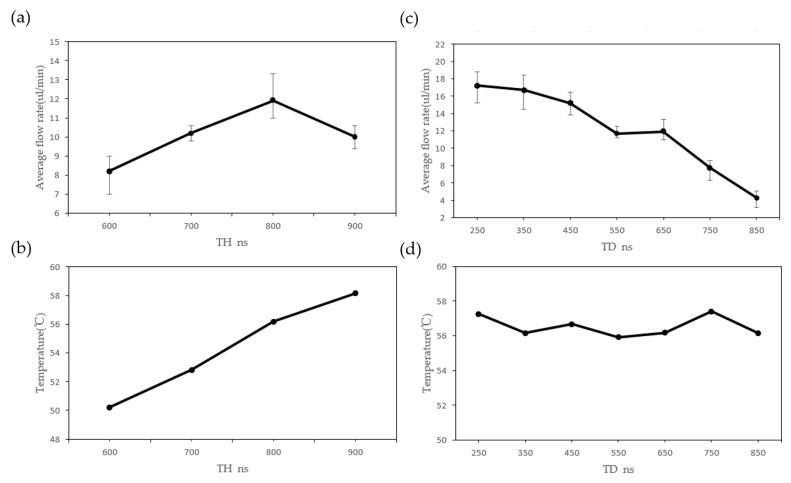
(**a**) The effects of TH on average flow rate; (**b**) the relationship between TH and temperature; (**c**) the effects of TD on average flow rate; and (**d**) the relationship between TD and temperature.

**Figure 14 sensors-23-05207-f014:**
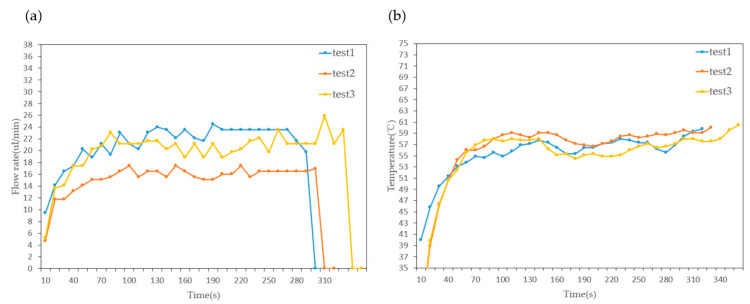
(**a**) Variation in flow rate over time and (**b**) variation in temperature over time.

**Figure 15 sensors-23-05207-f015:**
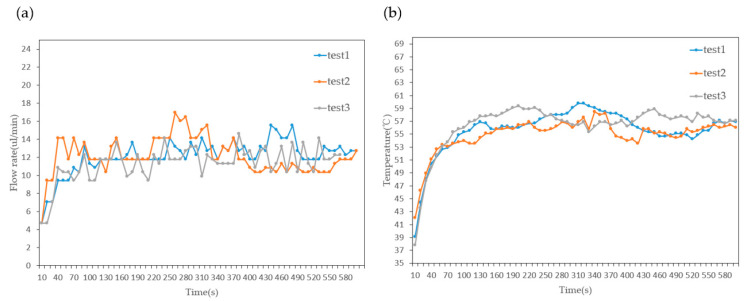
(**a**) Variation in flow rate over time and (**b**) variation in temperature over time.

**Table 1 sensors-23-05207-t001:** Comparison of the features of different micropumping techniques.

Parameter	Phase-Change Micropump [3]	Electrohydrodynamic Micropump	Electroosmotic Micropump [7]	Magnetohydrodynamic Micropump [7]
Bidirectional	Yes	Yes	Yes	Yes
Miniaturization potential	High	High	Medium	Medium
Moving parts	No	No	No	No
Fabrication (ease of alignment)	Silicon micromachining; heating alignment required	Silicon micromachining; no alignment	Not compatible with silicon micromachining	Silicon micromachining
Advantages and disadvantages	Simple structure; low cost	Large electric field close to the chip	Difficult to integrate into a pump	Low voltage; continuous flow

**Table 2 sensors-23-05207-t002:** Comparison with other published phase-change micropumps.

Parameter	Reference [18]	Reference [19]	Reference [7]	This Work
Cross-sectional area of the microchannel (µm^2^)	3.14 × 10^6^	1.256 × 10^5^	1.96 × 10^5^	1.2 × 10^3^
Length of microchannel (mm)	372	300	104	1.47
Flow rate (µL/min)	300	33	63	22
Control system	PLC	PLC	PLC	On-chip
Fabrication	Silicon micromachining	Silicon micromachining	Silicon micromachining	CMOS-MEMS

**Table 3 sensors-23-05207-t003:** Comparison with a passive phase-change micropump.

Parameter	Reference [21]	This Work
Insulating layer (µm)	1.5	3
Average flow rate (µL/min)	8.6	8.57
Printing frequency (kHz)	24.5	5
Coefficient of variation in the average flow rate	6%	3%
Temperature (°C)	66	47

**Table 4 sensors-23-05207-t004:** Comparison between constant heating conditions and optimized heating conditions.

Parameter	Constant Heating Conditions	Optimized Heating Conditions
Average flow rate (µL/min)	18.8	12
Coefficient of variation in the average flow rate	13%	3%
TH/TD (ns)	800/250	800–700–600–500/250
Printing frequency (kHz)	7	7
Temperature (°C)	58	57

## Data Availability

The data sets used or analyzed during the current study are available from the corresponding author upon reasonable request.

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
