# Peer review of "A Smart Active Phase-Change Micropump Based on CMOS-MEMS Technology"

_sensors, 2023, doi:10.3390/s23115207_

Round 1
Reviewer 1 Report
This manuscript proposed an active phase change micropump fabricated by CMOS-MEMS technology. The pumping mechanism is known already for a long time, but the novelty of the work is in the development of CMOS-MEMS integrated device. The study should be of interest to the readers of this journal. However, there are several issues that have to be sorted before publishing the manuscript.
Regarding the pumping mechanism, in case I, why is the fluid flowing from outlet to inlet when there is no pressure gradient? The pumping mechanism should be repetition of 1 then 2 then 3, so I think "step" or "phase" is more suitable than "case" to mention these.
Regarding the fabrication, Fig. 5 is quite difficult to understand. Please indicate in the figure, which part is what to assist the reader in understanding the optical micrograph. There is also a mismatch in the length of microchannel mentioned in Table 2 and narrative, which one is correct? The fabrication process also lacks some detail. Please mention which part of Fig. 5 (the second?) is under explanation in the narrative. What does it mean with the through silicon vias technology? And what material does each color in Fig. 5 (the second?) indicate? Please also reconfirm the figure numbering.
The authors used a ruler for measuring the average flow rate, which is quite traditional. How is the measurement accuracy, as the authors are working in μL order? How did the authors consider the error bars in Fig. 9(a)? An integrated flow sensor (10.1109/TIE.2020.2984446, 10.1109/JSEN.2021.3115656) should have been able to be incorporated in the chip, or a flexible flow sensor (10.3390/mi14010042, 10.1109/JSEN.2023.3272310) can also be attached at the outlet to more accurately measure the airflow rate. I suggest the authors to add the above literature in the manuscript as a reference for future studies. Such additions can also equip the pump with flow monitoring function.
The temperature reading in Fig. 9(b) shows a significantly lower temperature. What does this temperature actually indicate? Should we just see the increasing trend? What is the "insulating layer" referred to in the explanation around line 288? Thermal sensors (some of which I mentioned in the suggested references) are mostly fabricated as a freestanding or diaphragm structure to limit the heat loss to the substrate. However, I cannot see any such a structure in the present study. Please also provide scale bars for micrographs in Fig. 10.
Please elaborate about the coefficient of variation of average flow rate in Table 4. What does it mean?
Generally fine. But I would like to suggest a more extensive proof reading. For instance please check line 268.
Author Response
Thank you very much for the excellent comments.
We sincerely thank the reviewer for thoroughly examining our manuscript and providing very helpful comments to guide our revision.
Sincere thanks should be given to the reviewer for the constructive comments and suggestions. The responses to the comments are given in the attachment.

Reviewer 2 Report
The essay writing is logical and fluent. The design of the micropump is innovative and has been experimentally demonstrated. It has certain practicality in the field of micropump design.
Author Response
Thank you very much for the excellent comments.
We sincerely thank the reviewer for thoroughly examining our manuscript and providing very helpful comments to guide our revision.
Sincere thanks should be given to the reviewer for the constructive comments and suggestions.
Reviewer 3 Report
In present work, the authors describe an experimental and theoretical study on a CMOS-MEMS based active phase change micro-pump. They report a simplified model to analize the pumping effect, show the setup for the control of the micro-pump, and finally describe the optimization of the characteristic parameters.
The work is interesting and accurately described, and could be considered for the publication on Sensors.
Minor editing of English language are required
Author Response

(The authors gave the same response as above.)

Reviewer 4 Report
Authors present their design of a phase-change MEMS micropump.
In the Material section (2.1) it is not common or necessary to state from which supplier companies weas the equipment used for the research purchased from.
Also, type of certain equipment used for specific measurement type is usually mentioned in the body of text where experiments are described, otherwise it doesn't bear any information to the reader.
Section 2.2: Description of principle of operation is not clear, should be improved. Directions of flow (I-->O or O-->I) from the description of Case 1 and from the Fig. 1 do not match. Fig. 1 should be improved to more clearly show which heater is in operation in which case.
Description of temperature sensor design: it is not stated clearly enough stated that absolute temperature is implemented by measuring the BE junction-temperature of BJT transistors. Role of the remaining MOSFET transistors of the temperature measurement circuit from Fig. 4 is not described.
In addition to the temperature sensor, author should consider showing the design of major logic blocks from Fig. 3.
Fig. 5.a: Micrograph would be more descriptive if additional parts of the system apart from the microfluidic channel would be labeled.
Numbers on horizontal/vertical scales of figures showing results (e.g. Fig. 13, 14) are too small to read.
Paper contains plenty grammatical and orthographic errors which need to be corrected - English terminology, singular/plural, spaces missing, capital letters, writing signal and variable names etc. There are simply too many of them to list them all here, individually.
Author Response

(The authors gave the same response as above.)

Round 2
Reviewer 4 Report
Paper now seems fine.